# Progressive Pseudo Bag Augmentation with Instance Importance Estimation for Whole Slide Image Classification

## Abstract

In the field of computational pathology, the classification of whole-slide images (WSI) remains a challenging task due to the vast amount of gigapixel information and the limited availability of refined manual annotations. Recently, multiple instance learning (MIL) has emerged as a promising approach to address this issue. While attention-based MIL methods utilize attention mechanisms to distill instance information for training or further fine-tuning, the current ranking of attention scores fails to accurately locate positive instances. In this study, we propose the instance importance score (IIS) based on the Shapley value to tackle this problem. This approach enables the identification and prioritization of crucial features. Building upon this foundation, we present a novel framework for the progressive assignment of pseudo bags. Through comprehensive experiments, our approach achieves state-of-the-art performance compared to other superior methods on the CAMELYON-16, BRACS, and TCGA-LUNG datasets. Furthermore, the visualization results demonstrate the enhanced interpretability provided by the IIS in the classification of WSI. Code for our framework is accessible at https://github.com/*****.

## 1 Introduction

In recent years, computational pathology has undergone rapid advancements driven by the progress in digital imaging techniques. These advancements have transformed stained tissue specimens into comprehensive whole-slide images (WSIs), which serve as a basic resource for advanced diagnostic procedures. Deep learning-based computational algorithms, operating on patches tiled from WSIs, play a pivotal role in discerning essential features and making critical decisions across various clinical tasks Campanella et al. (2019); Yan et al. (2023). Among these tasks, WSI classification stands out as a significant endeavour, yet it confronts substantial challenges. The primary challenge lies in learning gigapixel-level information with only slide-level labels, as the refined manual annotations are prohibitively expensive Zhu et al. (2023); Chen et al. (2022); Yufei et al. (2022). Moreover, there is a growing demand from the clinical for an approach that delivers high performance and offers interpretability Pati et al. (2022); Jaume et al. (2021); Schwab & Karlen (2019).

To overcome these challenges, researchers have employed multiple instance learning (MIL), which is a weakly supervised approach that aggregates instances within a bag for classification. In its early stages, MIL approaches employed mean pooling or max pooling for feature aggregation Pinheiro & Collobert (2015); Feng & Zhou (2017); Zhu et al. (2017). Moreover, attention-based pooling has taken the forefront due to its effectiveness in amalgamating information Ilse et al. (2018). Recent studies have introduced methods that operate under the assumption that the ranking of attention scores can accurately identify positive instances. Lu et al. introduced an additional cluster branch founded on attention scores to distinguish features via projection Lu et al. (2021). Yu et al. proposed a Bayesian collaborative learning (BCL) framework, which assigns slide-level labels to patches garnering the highest attention for patch-level training, effectively fine-tuning the feature encoder by the agent task Yu et al. (2023). Li et al. identified and selected instances with top-ranking attention scores for end-to-end MIL training, addressing the information bottleneck Li et al. (2023). However, as depicted in Fig. 1, the attention distributions of different MIL models reveal that the top 5 instances collectively receive a significant portion of the attention scores. Moreover, when examining

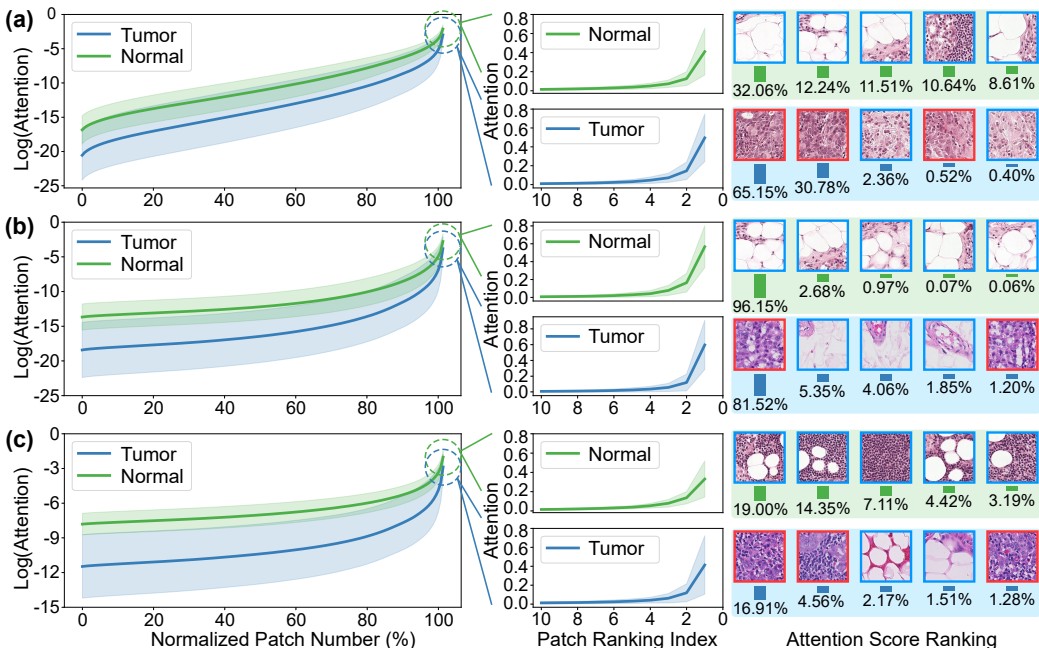

Figure 1: The attention distributions generated from attention-based pooling in CAMELYON-16, along with examples of the top 5 instances. (a), (b), and (c) employ ABMIL, CLAM, and DTFD as the backbone models, respectively. The attention scores from all patches across all slides are normalized for visualization, and the top 5 instances from one example slide are extracted and ordered.

the attention score rankings, it becomes apparent that positive instances can be misordered, even within the top 5 samples. This leads to the observation that attention can be fuzzy and deceptive, as it tends to concentrate on a limited subset of instances, resulting in a noisy ranking of instance importance rather than a precise one.

To promote MIL models to learn a greater number of positive instances, one approach is to partition the regular bag into multiple pseudo bags Shao et al. (2021a). Zhang et al. adopted a strategy where one bag is randomly partitioned into multiple pseudo bags for both training and inference Zhang et al. (2022). In order to solve the mislabeling issue of pseudo bags, they distilled a feature vector from each pseudo bag, and proposed a Tier-2 MIL model upon the distilled features for slide classification. However, as depicted in Fig.1 (c), their method does not fundamentally resolve the inherent mislabeling issue in pseudo bag assignment, which can adversely affect the MIL process.

Inspired by cooperative game theory Shapley et al. (1953), we introduce a metric to measure the contribution of each instance termed instance importance score (IIS). In cooperative game theory, the classical Shapley value serves as an indicator for comprehensively measuring the contribution of features under different cooperative relationships. The concept of the Shapley value can also be applied to the field of pathology WSI classification, where multiple patches contribute to the final diagnosis. Building upon the introduced IIS, we propose a MIL framework called PMIL, which incorporates progressive pseudo bag augmentation. This approach systematically divides a regular bag into a series of pseudo bags, enhancing the model's fitting and generalization capabilities. In summary, our key contributions are as follows:

- Acknowledging the limitations of attention scores in terms of ranking accuracy and interpretability, we tackle this issue by introducing the Shapley value-based IIS value as a measure of instance contributions in the context of multiple instance learning.

- We propose a framework, called PMIL, that utilizes IIS to gradually assign pseudo bags, effectively enhancing the MIL model.

- Extensive experiments have been conducted, which demonstrates that our method achieves a state-of-the-art level of performance and provides improved interpretation.

## 2 METHOD

### 2.1 PSEUDO BAG AUGMENTED MULTIPLE INSTANCE LEARNING FOR WSI CLASSIFICATION

To combat the challenges in whole-slide image classification, we first retrospect the pseudo bag augmented multiple instance learning. We denote the training set of labeled WSIs as $\mathcal{D} = \{X_i, Y_i\}_{i=1}^{|\mathcal{D}|}$, where $X_i = \{x_{i,j}\}_{j=1}^{N_i}$ represents the $i$th bag (slide) of $N_i$ instances after feature extraction, and $|\mathcal{D}|$ is the number of labeled bags. Traditional MIL involves aggregating instances into a bag-level representation and mapping it to a bag-level prediction, as follows:

$$\hat{Y} = f\left(g\left(\{x_{i,j}\}_{j=1}^{N_i}\right)\right), \tag{1}$$

where $g(\cdot)$ is the aggregator and $f(\cdot)$ is the fully connected (FC) layer. Considering randomly splitting a regular bag into $M$ pseudo bags, each pseudo bag inherits the label from its parent bag, resulting in an expanded training set $\mathcal{D}^{pse} = \{X_i^{pse}, Y_i\}_{i=1}^{M \times |\mathcal{D}|}$. We can obtain $\hat{Y}^{pse}$ via Eq.1, and the objective function for pseudo bag augmented MIL is then defined as,

$$\mathcal{J}\left(\mathcal{D}^{pse}; \theta\right) = \sum_{i=1}^{M \times |\mathcal{D}|} \mathcal{L}\left(\hat{Y}_i^{pse}, Y_i\right), \tag{2}$$

where $\theta$ is the parameter of the MIL classifier, including the aggregator and the FC layer. The loss function $\mathcal{L}$ used in this work is the cross entropy loss. However, this augmentation can assign pseudo bags with incorrect labels. The objection function in Eq.2 can be further divided into two items:

$$\mathcal{J}\left(\mathcal{D}^{pse}; \theta; \varepsilon\right) = \sum_{i=1}^{M \times |\mathcal{D}| - \varepsilon} \mathcal{L}\left(\hat{Y}_i^{pse}, Y_i \middle| Y_i = Y_i^{pse}\right) + \sum_{i=1}^{\varepsilon} \mathcal{L}\left(\hat{Y}_i^{pse}, Y_i \middle| Y_i \neq Y_i^{pse}\right), \tag{3}$$

where $\varepsilon$ represents the number of pseudo bags with incorrectly assigned labels. Eq.3 reveals a trade-off in the training process, dependent on $\varepsilon$: the first term increases the number of training bags, thereby bolstering the diversity of positive instances; while the second term introduces training noise, leading to an unstable training. A common practice is to randomly split pseudo bags, which fixes $\varepsilon$ for optimization. Our target is to obtain proper $\theta$ and $\varepsilon$ to minimize the overall objection function, which can be decoupled into optimizations of $\theta$ in Eq.2 and $\varepsilon$ by approximation. However, we can not directly improve $\varepsilon$ since the true label of the pseudo bag is not available. Thus, we transfer this issue to the optimization of pseudo bag assignment.

### 2.2 SHAPLEY VALUE-BASED INSTANCE IMPORTANCE SCORE ESTIMATION

To fully leverage the benefits of pseudo bag augmentation, we introduce the concept of instance importance scores (IIS) to estimate the contribution of each instance, guiding the process of splitting regular bags into pseudo bags to minimize $\varepsilon$.

In attention-based MIL models, using attention scores derived from pooling operations as IIS is a logical choice. As shown in Fig.1, our observation reveals that attention scores might not accurately reflect the ranking of importance, as attention-based pooling often prioritizes a small subset of instances. To address this limitation, we introduce the Shapley value as an alternative method for estimating IIS. In the context of the MIL framework, this approach necessitates evaluating the model across all feasible instance subsets from the full set of instances $S_i \subseteq X_i$. The Shapley value $\phi$ for a particular instance $x_{i,j}$ is calculated by considering the differences in model predictions with and without $x_{i,j}$ for all feature subsets $S_i \subseteq X_i \backslash \{x_{i,j}\}$:

$$\phi_{i,j} \triangleq \sum_{S_i \subseteq X_i \backslash \{x_{i,j}\}} \frac{|S_i|! \left(|X_i| - |S_i| - 1\right)!}{|X_i|!} \left[f\left(g\left(S_i \cup \{x_{i,j}\}\right)\right) - f\left(g\left(S_i\right)\right)\right]. \tag{4}$$

The computation of Shapley value takes a comprehensive consideration on the contribution of each instance. While directly calculating the Shapley value for computational pathology, where one bag often contains thousands of instances, can be time-consuming. To expedite the computation, several methods have been proposed to approximate Shapley values via sampling Strumbelj & Kononenko

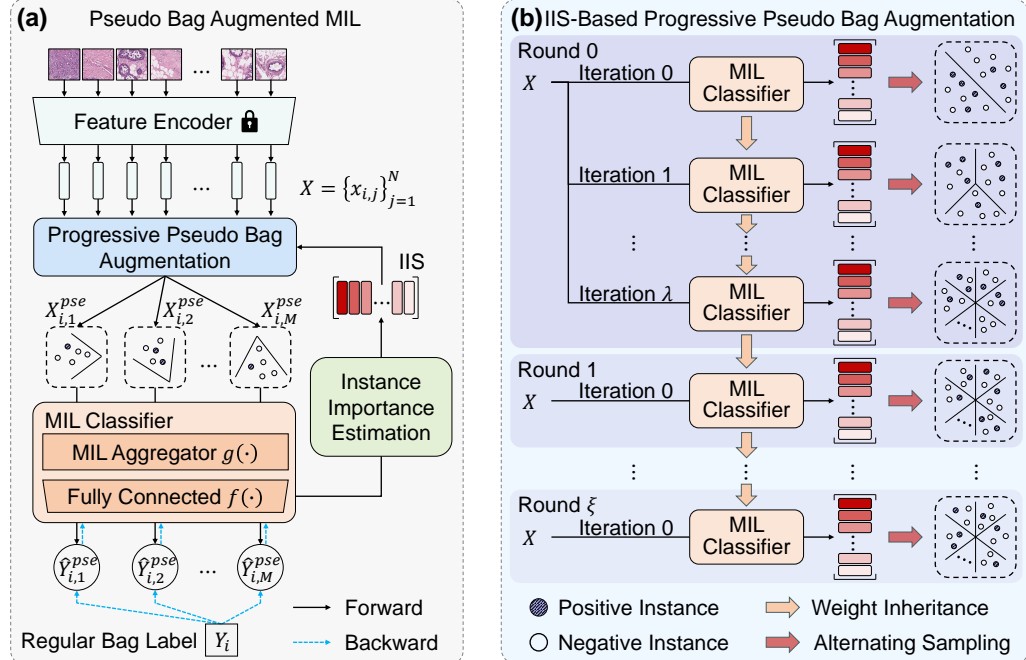

Figure 2: Overview of the proposed PMIL framework during the training process. (a) Initialize the feature encoder with pretrained parameters and set frozen, then assign $M$ pseudo bags based on the calculated instance importance score (IIS), and train the pseudo bag augmented MIL model with the regular bag label. (b) Progressively increase the number of pseudo bags and improve the pseudo bag initialization across various training iterations and rounds.

(2010), weighted regression, a modified backpropagation step Lundberg & Lee (2017), and other approaches Ancona et al. (2019); Chen et al. (2018). These methods aim to reduce the computational complexity by minimizing the number of subsets required for each instance. It's worth noting that the crucial instances are typically the positive ones, and their order significantly impacts the accuracy of labels assigned to pseudo bags. To accelerate the computation, we focus on instances with high attention scores, which are more likely to significantly influence the final prediction, and leave the less significant instances to serve as the subset range for sampling. Thus, the Shapley value of instances with high attention scores $\phi_{i,j} | x_{i,j} \in S_i^h$ can be accelerated as,

$$\text{IIS}\left(x_{i,j}\right) = \sum_{S_{i,j} \subseteq S_i^l, j=1}^{k} \frac{|S_{i,j}|! \left(|S_i^l| - |S_{i,j}| - 1\right)!}{|S_i^l|!} \left[f\left(g\left(S_{i,j} \cup \{x_{i,j}\}\right)\right) - f\left(g\left(S_{i,j}\right)\right)\right], \quad (5)$$

where $S_i^h$ is the instance subset with high attention scores, $S_i^l = X_i - S_i^h$ is the complementary set of $S_i^h$ in the case where $X_i$ is a universal set. The instance number of $S_i^h$ is set to $\mu M$, where $\mu$ is set to 10 in this work, and the sampling number $\tau$ for $S_i^l$ is set to 3 for further experiments.

Assuming that the computational time of reasoning once in the MIL model per bag is a constant $\gamma$, we can quantify the computational complexity of different IIS calculation methods as follows:

$$\Omega\left(\text{Shapley Value}\right) = \sum_{i=1}^{|\mathcal{D}|} \sum_{j=0}^{N_i} C_{N_i}^j \cdot \gamma = \gamma \sum_{i=1}^{|\mathcal{D}|} 2^{N_i}, \quad (6)$$

$$\Omega\left(\text{Accelerated Shapley Value}\right) = \sum_{i=1}^{|S_i^h|} \sum_{j=0}^{\tau} \gamma = \gamma \tau \mu M. \quad (7)$$

where $C_{N_i}^j$ is combination number. Through simplification, we can obtain an acceleration ratio of $\frac{\sum_{i=1}^{|\mathcal{D}|} 2^{N_i}}{\tau \mu M}$ on the Shapley value as our proposed IIS, ensuring a linear computational complexity while maintaining ranking accuracy.

## 2.3 Instance Importance Score-Based Progressive Pseudo Bag Augmentation

Based on the calculated instance importance score, we can iteratively reorder instances within each bag $X_i' = \left\{ x_{i,j}' \mid \text{IIS}\left(x_{i,1}'\right) \geq \text{IIS}\left(x_{i,2}'\right) \geq \cdots \geq \text{IIS}\left(x_{i,N_i}'\right)\right\}$, as illustrated in Fig. 2. These reordered instances are interleaved into $M$ pseudo bags, namely, each pseudo bag $X_{i,k}^{pse} = \left\{ x_{i,j}' \mid j \equiv k \,(mod\, M)\right\}$ contains instances alternating sampled from $X_i'$. Thus, the optimization of $\varepsilon$ can be approximated as:

$$\varepsilon^* = \arg\min_{X'} D_{KL}\left(P_{X^{pse} \sim \Gamma(X')}\left[Y^{pse} \mid X^{pse}; \theta\right] \parallel P\left[Y \mid X; \theta\right]\right), \tag{8}$$

where $\Gamma\left(X'\right)$ is the instance importance distribution of $X'$. It is well-established that the optimization defined by Eq.2 and Eq.8 can be solved by using the EM algorithm. By optimizing $\Gamma\left(X'\right)$, each pseudo bag is more likely to contain at least one positive instance to minimize $\varepsilon$, namely the risk of mislabeling. In this work, only bags within the training set are split into pseudo bags, and the training process for these pseudo bags remains identical to that of the regular bags.

Directly splitting a regular bag into a large amount of pseudo bags can introduce training instability, particularly when the MIL model struggles to capture crucial instances. This instability is especially problematic when a regular bag contains only a few positive instances, such as in the case of micro metastasis. To address this issue, we progressively increase the number of pseudo bags during the early training process when the MIL model converges to a locally optimal solution, denoted by:

$$M_t = \min\left\{M_{t-1} + \Delta M, M_{max}\right\}, s.t. \left\{g_{t-1}, f_{t-1}\right\} \to \left\{g_{t-1}^*, f_{t-1}^*\right\}, \tag{9}$$

where $t$ is the convergence iteration, $\Delta M$ is the pseudo bag number increment, and $M_0$ and $M_{max}$ are the initial and maximum pseudo bag numbers, respectively.

The initial assignment of instances to pseudo bags plays a crucial role in the subsequent training, especially when dealing with challenging datasets. To address this issue, we introduce additional EM training rounds to optimize the MIL model and the distribution of $X'$. In each round $\xi$, we progressively enhance the initial pseudo bag augmentation by calculating the initial instance importance scores at the first iteration, using the well-trained MIL model from the previous round $\xi - 1$.

## 3 Experiments

### 3.1 Datasets and Evaluation Metrics

In our experiments, we report one-versus-rest area under curve (AUC), slide-level accuracy (ACC), and macro F1 score as the evaluation metrics. We utilize three public pathology WSI datasets to assess our methods.

CAMELYON-16 is designed to detect lymph node metastasis in early-stage breast cancer. It comprises 399 WSIs, with 270 allocated for training and 129 for testing. The official training set follows a 5-fold cross-validation protocol to generate training and validation sets. We report the mean performance metrics on the official test set.

BRACS Brancati et al. (2022) is curated for breast cancer subtyping and contains 547 H&E-stained WSIs. The classification task involves benign tumors, atypical tumors (AT), and malignant tumors (MT). We adhere to the official dataset split, with 395 for training, 65 for validating, and 87 for testing. We conduct five separate experiments with different random seeds and report the mean performance metrics on the official test set.

TCGA-LUNG comprises 1034 WSIs, encompassing 528 lung adenocarcinoma (LUAD) and 506 lung squamous cell carcinoma (LUSC) cases. We adopt a 5-fold cross-validation protocol for both training and testing. The mean performance metrics are reported on the test set.

In our preprocessing step, we employ OTSU's threshold method to localize tissue regions for patch generation. Non-overlapping patches of size 256×256 pixels are tiled at a 20× magnification for CAMELYON-16 and TCGA-LUNG, and a 5× magnification for BRACS, yielding an average of about 7156, 11951, and 714 patches per bag, respectively.

Table 1: Results on CAMELYON-16, BRACS, and TCGA-LUNG test set. The encoder ResNet50 is pretrained on ImageNet. The subscripts are the standard variances. The best evaluation metrics are in bold.

| Method | CAMELYON-16 | | | BRACS | | | TCGA-LUNG | | |
|---|---|---|---|---|---|---|---|---|---|
| | ACC | AUC | F1 | ACC | AUC | F1 | ACC | AUC | F1 |
| MeanMIL | $70.9_{1.8}$ | $58.7_{1.9}$ | $62.3_{3.3}$ | $52.4_{2.6}$ | $69.2_{1.6}$ | $40.6_{2.5}$ | $82.0_{0.9}$ | $88.9_{2.0}$ | $82.0_{1.0}$ |
| MaxMIL | $83.7_{1.8}$ | $86.7_{2.6}$ | $83.3_{5.1}$ | $55.9_{2.8}$ | $75.9_{1.6}$ | $50.3_{4.0}$ | $88.7_{1.0}$ | $94.4_{1.2}$ | $88.7_{1.0}$ |
| ABMIL | $82.5_{1.9}$ | $83.8_{2.1}$ | $80.6_{1.7}$ | $58.4_{0.9}$ | $76.1_{0.6}$ | $54.7_{2.3}$ | $87.6_{0.7}$ | $93.1_{1.8}$ | $87.6_{0.7}$ |
| DSMIL | $77.2_{1.7}$ | $77.2_{2.1}$ | $74.4_{2.6}$ | $53.1_{2.2}$ | $70.8_{3.3}$ | $46.1_{3.7}$ | $86.2_{1.4}$ | $93.6_{1.0}$ | $86.2_{1.4}$ |
| CLAM | $82.5_{3.2}$ | $81.6_{2.4}$ | $80.1_{3.5}$ | $53.8_{3.5}$ | $73.3_{1.7}$ | $51.5_{3.3}$ | $88.2_{1.4}$ | $94.2_{1.2}$ | $88.2_{1.4}$ |
| TransMIL | $85.0_{1.4}$ | $89.1_{0.7}$ | $83.3_{1.3}$ | $57.0_{2.4}$ | $75.5_{1.0}$ | $49.2_{5.2}$ | $87.9_{0.9}$ | $94.8_{0.8}$ | $87.9_{0.9}$ |
| DTFD | $85.3_{1.6}$ | $85.4_{3.2}$ | $84.9_{1.7}$ | $57.2_{2.7}$ | $76.6_{2.0}$ | $56.2_{3.8}$ | $88.8_{0.6}$ | $94.6_{0.8}$ | $88.8_{0.6}$ |
| PMIL | $\mathbf{87.4}_{1.1}$ | $\mathbf{90.1}_{1.6}$ | $\mathbf{86.3}_{1.1}$ | $\mathbf{68.3}_{1.7}$ | $\mathbf{84.0}_{0.3}$ | $\mathbf{66.5}_{2.4}$ | $\mathbf{91.3}_{1.4}$ | $\mathbf{96.5}_{0.9}$ | $\mathbf{91.3}_{1.4}$ |

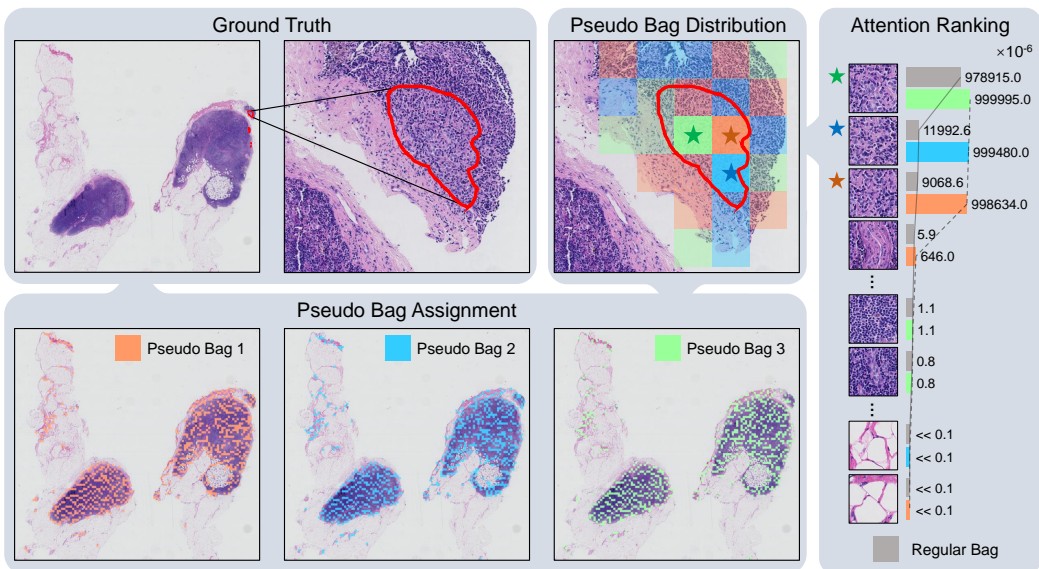

Figure 3: Visualization of pseudo bag assignment using PMIL. The red annotations represent cancer regions. The five-pointed star pointed patches obtained the most attention in each pseudo bag.

## 3.2 IMPLEMENTATION DETAILS

All experiments were conducted on a workstation equipped with NVIDIA RTX 3090 GPUs. For model training, we used ResNet50 as the encoder He et al. (2016), and ABMIL as the backbone MIL model. We employed the Adam optimizer with a weight decay of 1e-5, and implemented the early stopping strategy with a patience setting of 20 epochs. The initial learning rate was set to 3e-4 and reduced to 1e-4 for finetuning. We set the maximum number of pseudo bags to 8 for the CAMELYON-16, 10 for the BRACS, and 14 for TCGA-LUNG. Furthermore, the number of pseudo bags gradually increased by 4 with the first training round during the training process.

## 3.3 PERFORMANCE COMPARISON

We present the experimental results of our proposed methods on CAMELYON-16, BRACS, and TCGA-LUNG datasets. These results are compared to those obtained by the following MIL methods: Mean-Pooling, Max-Pooling, the classic AB-MIL Ilse et al. (2018), DSMIL Li et al. (2021), CLAM-SB Lu et al. (2021), TransMIL Shao et al. (2021b), and DTFD Zhang et al. (2022).

As illustrated in Tab. 1, our proposed PMIL method stands out with impressive AUC scores of 90.1% for CAMELYON-16, 84.0% for BRACS, and 95.6% for TCGA-LUNG, consistently surpassing

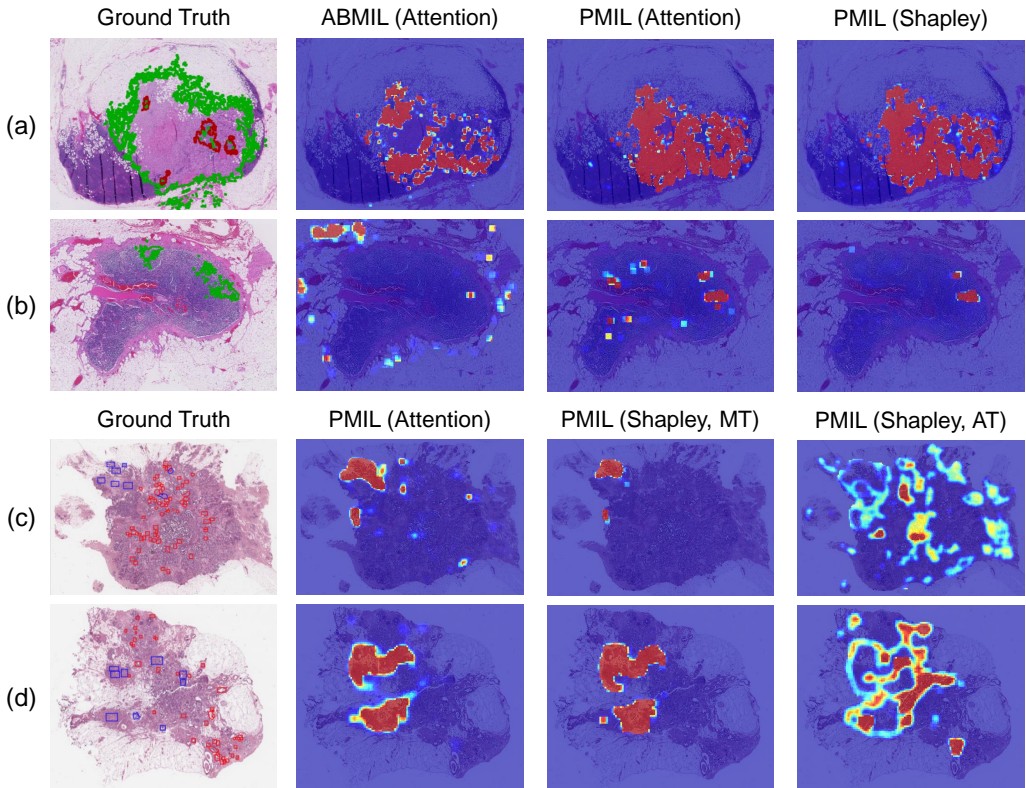

Figure 4: Visualization of the heatmaps of different models and IIS on the CAMELYON16 and BRACS datasets. (a) and (b) are macro and micro metastasis cases from CAMELYON-16, where the green annotations represent cancer regions, and the red annotations represent noncancer regions. (c) and (d) are malignant tumor cases from BRACS, where blue annotations represent the malignant tumor (MT) regions, and the red annotations represent the atypical tumor (AT) regions.

all other methods in the comparison. Especially in the challenging BRACS dataset, our method outperforms other methods significantly. By progressively generating reasonable pseudo bags, we enhance training diversity and reduce the number of instances in each bag, ultimately facilitating the model's ability to learn positive instances effectively.

## 3.4 VISUALIZATION AND INTERPRETATION

To illustrate the accuracy of our pseudo bag augmentation, as depicted in Fig. 3, all patches have been segmented into three pseudo bags based on the calculated Shapley values. Notably, the three critical patches within the micro metastasis region are evenly distributed among different pseudo bags, which demonstrates our approach enhances the diversity of positive instances.

To emphasize the limitations of the attention score, we conducted a comparative analysis, as illustrated in Fig. 4 (a) and (b). All model performs well in the macro metastasis case. However, in the case of micro metastasis, the calculated attention score indicates that ABMIL and our model focus on some noncancerous areas. However, when employing the Shapley value, our proposed model accurately excludes the negative regions and precisely identifies the cancer regions.

Unlike the attention value, the Shapley value computation utilizes the entire MIL classifier, inherently containing category information. As illustrated in Fig. 4 (c) and (d), both attention score and Shapley value (MT) predominantly concentrate on malignant tumor regions, aligning with the slide-level labels. However, when setting atypical tumors as the category for Shapley value, the heatmaps primarily highlight the atypical tumor regions. Although the heatmaps may not achieve pinpoint accuracy on the BRACS dataset due to limited performance, this observation underscores the robust interpretability of the Shapley value in multi-classification tasks. These visualization results reveal

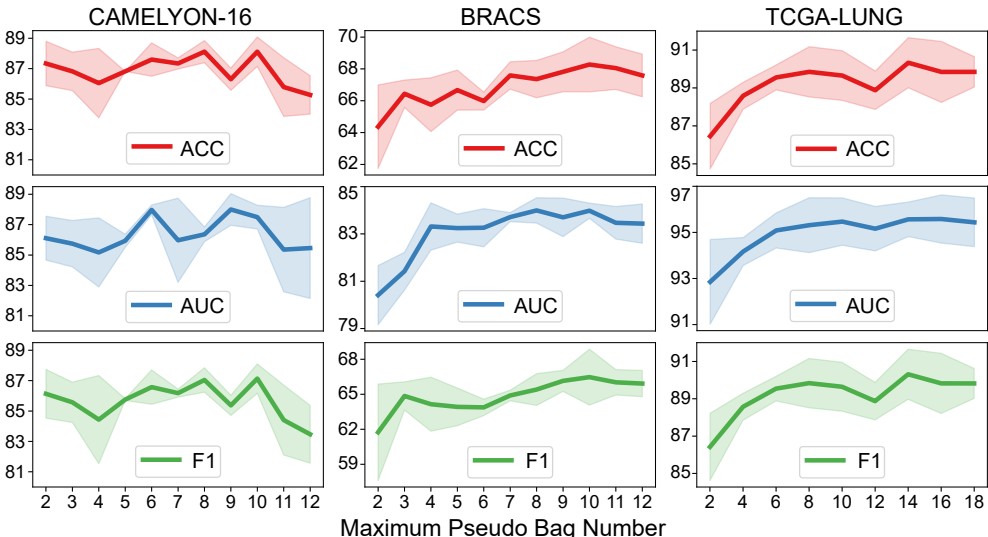

Figure 5: Performances of different maximum pseudo bag numbers $M_{max}$ on three datasets.

that the attention score often captures a noisy ranking of instance importance, whereas the Shapley value effectively addresses this issue, resulting in a more accurate and reliable interpretation. Furthermore, while the attention score only provides insights into the heatmaps of the target category, the Shapley value has the capability to highlight additional category information.

## 3.5 ABLATION STUDY

### 3.5.1 IIS MEASURE METRICS

We employed both the attention score and the Shapley value to measure the IIS for further training, and we used random split as the baseline pseudo bag augmentation strategy. The results in Tab. 2 indicate that the Shapley value-based method outperformed others on the CAMELYON-16 and TCGA-LUNG datasets. In contrast, the attention score-based method performed better on the BRACS dataset. This difference in performance can be attributed to the fact that attention scores can be directly obtained through pooling operations, while the Shapley value computation involves an additional fully connected (FC) layer. The Shapley value may exhibit reduced robustness when the accuracy of the MIL classifier is not sufficiently high; however, it tends to be a better choice when dealing with datasets that are less challenging to learn.

### 3.5.2 MAXIMUM PSEUDO BAG NUMBER

The choice of the maximum pseudo bag number varies depending on the dataset due to differences in magnification levels of the tiled patches and the size of tumor regions. As illustrated in Fig. 5, our model achieves the best performance on the CAMELYON-16 dataset when the maximum pseudo bag number is set to 8 or 10. However, exceeding this range leads to a sharp decline in performance. In contrast, for the BRACS and TCGA-LUNG datasets, our model performs better as the maximum pseudo bag number increases and achieves the best performance when the number is set to 10 and 14, respectively. This is because the CAMELYON-16 dataset contains abundant micro metastasis slides with only a few positive instances, even when tiled at a $20\times$ resolution. In this case, the augmentation faces a trade-off between adding more noise to the training set or enhancing training diversity. Meanwhile, the cancer (subtype) regions in the BRACS and TCGA-LUNG datasets are much larger and can be effectively divided into numerous pseudo bags.

## 3.6 PROGRESSIVE PSEUDO BAG AUGMENTATION STRATEGIES

To demonstrate the effectiveness of our progressive augmentation, we conducted experiments with different pseudo bag augmentation strategies. In these experiments, we utilized a pseudo bag num-

Table 2: Evaluation of pseudo bag assignment using different IIS measure metrics on CAMELYON-16, BRACS and TCGA-LUNG test sets. The best evaluation metrics are in bold.

| Metrics | CAMELYON-16 | | | BRACS | | | TCGA-LUNG | | |
|---|---|---|---|---|---|---|---|---|---|
| | ACC | AUC | F1 | ACC | AUC | F1 | ACC | AUC | F1 |
| Random | 87.13 | 86.19 | 85.49 | 61.78 | 80.39 | 58.46 | 89.65 | 95.77 | 89.64 |
| Attention Score | **87.44** | 89.76 | 86.28 | **68.28** | **83.98** | **66.47** | 90.33 | 95.57 | 90.31 |
| Shapley Value | **87.44** | **90.10** | **86.30** | 66.09 | 81.27 | 64.67 | **91.29** | **96.45** | **91.29** |

Table 3: Evaluation of pseudo bag assignment using different progressive strategies on CAMELYON-16, BRACS and TCGA-LUNG test sets. The best evaluation metrics are in bold.

| Pseudo Bag Strategy | | CAMELYON-16 | | | BRACS | | | TCGA-LUNG | | |
|---|---|---|---|---|---|---|---|---|---|---|
| Number | Initialization | ACC | AUC | F1 | ACC | AUC | F1 | ACC | AUC | F1 |
| Constant | Constant | 80.8 | 77.5 | 76.9 | 62.6 | 82.8 | 60.9 | 90.6 | 95.9 | 90.6 |
| Progressive | Constant | 84.9 | 85.5 | 82.4 | 64.7 | 82.2 | 62.1 | 89.7 | 95.6 | 89.7 |
| Constant | Progressive | 85.1 | 86.4 | 83.3 | 70.7 | 84.5 | 68.7 | 90.8 | 96.8 | 90.8 |
| Progressive | Progressive | **88.2** | **88.1** | **87.0** | **71.3** | **84.9** | **69.9** | **91.3** | **96.1** | **91.3** |

ber increment of 4 and set the training rounds to 5 for CAMELYON-16 and TCGA-LUNG, and 10 for BRACS, as this dataset is more challenging to learn. As depicted in Tab. 3, the model equipped with the full progressive strategy achieves the best performance. Progressively increasing the number of pseudo bags significantly impacts CAMELYON-16, as it requires fine adjustment to avoid introducing excessive noise. Meanwhile, progressively inheriting the initial weight from the previous round significantly influences the performance on BRACS, where subtypes are inherently difficult to discern. A good pseudo bag initialization facilitates the model's ability to locate positive instances, resulting in improved performance. In contrast, TCGA-LUNG is less difficult to learn, leading to an insignificant increase in performance.

From these ablation studies, we summarize several key insights as follows:

**Choice of IIS Metrics**. The variation of IIS metrics depends on the specific dataset characteristics. The attention score-based IIS is a reliable and commonly used choice. However, the Shapley value-based IIS performs better on relatively more accessible datasets but may exhibit reduced performance on more challenging datasets, we believe it's primarily due to its reliance on accurate classification results.

**Sensitivity to Maximum Pseudo Bag Number**. The $M_{max}$ is highly sensitive to different datasets according to the number of positive instances. In case of large tumor regions within bags, it is advisable to use a larger $M_{max}$. Conversely, for datasets with only a few positive instances per bag, a more conservative approach is to set a smaller $M_{max}$.

**Progressive Strategies**. Progressive increase in the number of pseudo bags is particularly effective for challenging datasets or those with only a limited number of positive instances in each bag. While this approach may be less appealing for datasets with substantial tumor regions. In contrast, progressive initialization represents a significant improvement across various datasets, especially on more challenging ones.

## 4 CONCLUSION

In this paper, we initially reveal the distribution and ranking problems associated with the attention score, which can result in suboptimal training and limited interpretability. To tackle these issues, we introduce the Shapley value-based IIS to measure the contribution of each instance, which guides a more rational assignment of pseudo bags. Furthermore, we propose a novel framework called PMIL, which leverages IIS to progressively assign pseudo bags. Through comprehensive experiments, our approach outperforms state-of-the-art methods on three public datasets and the Shapley value-based IIS provides enhanced interpretability for pathological whole slide images.

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
