# OpenReview forum: "Progressive Pseudo Bag Augmentation with Instance Importance Estimation for Whole Slide Image Classification"
_ICLR.cc/2024/Conference — ICLR 2024 Conference Withdrawn Submission_

### Official Review · Reviewer_vsHj · 2023-10-21

**Soundness:** 2 fair
**Presentation:** 2 fair
**Contribution:** 1 poor
**Rating:** 3
**Confidence:** 4

**Summary:**

This work proposes PMIL - a novel MIL framework. PMIL aims to address the limitations of using attention values in terms of ranking accuracy and interpretability. PMIL divides a bag into pseudo bags with a progressive schedule. Experiments on 3 MIL problems show improved results and interpretability.

**Strengths:**

1. The problem of interpretability in MIL for pathology is an important one, and this paper aims to tackle that through initializing pseudo bags based on IIS and a progressive increase in their number. Using the progressive strategy in particular is a smart way to take inspiration from the literature around imbalanced classification.
2. The experimental methodology is rigorous with a 5 fold CV performed over all 3 datasets. This has been missing in some previous SOTA works for MIL.
3. Writing style is crisp and clear.

**Weaknesses:**

1. My main question is around the proposed novelty - Additive MIL [1] proposes a MIL formulation based on Shapely values where the instances' contribution to the class can be added to obtain the bag-level contribution. This work doesn't mention comparisons with it and how it improves over that technique.
2. All the problems picked here are binary classification problems, how does this framework generalize to multi-class problems such as RCC subtyping in the TCGA dataset?
3. The potential reduced performance of the IIS score on challenging datasets and sensitivity to params like maximum pseudo bag number might indicate that it might require relatively more hyperparameter tuning and prior knowledge about the problem for newer datasets.


[1] - "Additive MIL: Intrinsically Interpretable Multiple Instance Learning for Pathology", NeurIPS 2023.

**Questions:**

- Can the authors conduct a qualitative study with a domain expert where they rank the heatmaps produced by this technique vs others for WSIs containing positive as well as negative labels?
- While a thorough ablation for M (pseudo bag number) is provided, what is the impact of instance_number and sampling_number (mentioned below eq 5 in section 2.2) on the performance, and how were these values decided?

---

### Official Review · Reviewer_WwcE · 2023-10-31

**Soundness:** 2 fair
**Presentation:** 3 good
**Contribution:** 1 poor
**Rating:** 3
**Confidence:** 3

**Summary:**

Attention-based MIL methods utilize attention mechanisms to distill instance information for training or further fine-tuning, the current ranking of attention scores fails to accurately locate positive instances. The authors propose the instance importance score (IIS) based on Shapley value to tackle this problem. They further present a framework for the progressive assignment of pseudo bags.

**Strengths:**

The authors maintain a high-quality presentation and the motivation is clearly stated.

**Weaknesses:**

(1) One published work [1] based on Shapley value is not disscussed and compared in the paper.

(2) Recent models [2,3] that achieve strong performance are not compared in the experiments.

references:

[1] Javed, Syed Ashar, Dinkar Juyal, Harshith Padigela, Amaro Taylor-Weiner, Limin Yu, and Aaditya Prakash. "Additive MIL: intrinsically interpretable multiple instance learning for pathology." Advances in Neural Information Processing Systems 35 (2022): 20689-20702.

[2] Wang, Xiyue, Jinxi Xiang, Jun Zhang, Sen Yang, Zhongyi Yang, Ming-Hui Wang, Jing Zhang, Wei Yang, Junzhou Huang, and Xiao Han. "SCL-WC: Cross-slide contrastive learning for weakly-supervised whole-slide image classification." Advances in neural information processing systems 35 (2022): 18009-18021.

[3] Xiang, Jinxi, and Jun Zhang. "Exploring low-rank property in multiple instance learning for whole slide image classification." In The Eleventh International Conference on Learning Representations. 2022

**Questions:**

Could the authors explain why the related models are not included in the paper? I am happy to discuss further.

---

### Official Review · Reviewer_znAR · 2023-11-01

**Soundness:** 3 good
**Presentation:** 3 good
**Contribution:** 3 good
**Rating:** 8
**Confidence:** 4

**Summary:**

The authors frame their work around the limitations of attention vector in the aggregation function in a multiple instance learning framework for whole slide image (WSI) classification.  This formulation implements pseudobags where M subset of tiles inherit the bag level label for training. The clever augmentation to this frame work is the use of the so called instance importance score (IIS), which is based on the Shapley value where tiles are removed and the relative loss of predictive power by removal of important tiles is measure.   To improve efficiency the authors  only tiles with relatively high attention scores were evaluated for the IIS. With IIS scores for these patches ranked, pseudobags can be seeded sequentially with high IIS scores making it more likely that each pseudobag will have positive tiles if the initial bag is in fact positive. This process is done iteratively to optimize the number of pseudobags (M).  Using this method shows modest improvement of performance on benchmarking datasets. Some introspection is demonstrated, including the relative performance of the model on the three datasets with different level of maxM.

**Strengths:**

The approach is novel and is a creative augmentation to state of the art methods.  I have found that Shapley-based importance metrics are much better at providing some explainability to models than some other methods so seeing this applied on the aggregation function in MIL strikes me as a potential step in the right direction.

It is important to note that the attention weightings from aggregation function is useful but does consistently have confusing and spurious results if overly relied on.  An improvement on identification of the most important tiles for classification can potentially result in better whole slide classification and better explainability.

The submission is written very clearly and weaknesses are appropriately acknowledged.

**Weaknesses:**

In clinical practice the smaller the areas of tumor the more challenging the case is so degradation in performance in the datasets with smaller number of positive tiles may make it so this method is not the best choice for more challenging datasets.  This is well-acknowledged by the authors: "The Shapley value may exhibit reduced robustness when the accuracy of the MIL classifier is not sufficiently high; however, it tends to be a better choice when dealing with datasets that are less challenging to learn."

**Questions:**

There is an emphasis on the consideration that some of the tiles with high attention score are not positive for cancer visually.  How can you be certain that these tiles are not important for the prediction task in a way that is not obvious? For example there may be some stromal response that indicates cancer is present but not in that specific tile.

Can you regenerate Fig1 and demonstrate that rank of IIS does enrich better for tumor tiles?

Can you double check the legend for figure 4 is more clear.  It appears on a) that the ground truth image that green is not tumor or is perhaps the outline of the tumor.  The attention maps (red) appear to be labeling tumor correctly.  But legend does not clearly articulate what the colors mean in the attention/IIS heatmaps.